# Design of X-Concentric Braced Steel Frame Systems Using an Equivalent Stiffness in a Modal Elastic Analysis

**Claudio Amadio, Luca Bomben *** and Salvatore Noè

Department of Engineering and Architecture, University of Trieste, Piazzale Europa 1, 34127 Trieste, Italy; amadio@units.it (C.A.); noe@units.it (S.N.)
* Correspondence: luca.bomben@phd.units.it

**Abstract:** In this work, a general method for the design of concentric braced steel frames (CBF) with active tension diagonal bracings, applicable to single- and multi-storey structures, is presented. The method is based on the use of an elastic modal analysis with a response spectrum, which is carried out using an appropriate modified elastic stiffness of diagonal bracings. The reliability of the proposed method is validated through the analysis of significant case studies, making a series of numerical comparisons carrying out time-history non-linear dynamic analysis.

**Keywords:** concentric braced steel frames; active tension diagonal bracings; pushover analysis; seismic behaviour; dynamic time-history analysis

## 1. Introduction

Concentric braced steel frames with active tension diagonal bracings (X-CBF) represent one of the most commonly used structural types to withstand earthquakes or wind forces. They are widely used for both mono- and multi-storey buildings due to their high dissipative capacity and cheapness. During a seismic event, the energy is dissipated through the diagonals, which plasticize in tension and buckle in compression. The other parts, like beams and columns, are generally designed to remain elastic [1–3].

Many experimental and numerical studies, such as [4–6], have demonstrated that an X-CBF subjected to increasing horizontal actions is characterized by three-phase behaviour (Figure 1). In the first phase (the pre-buckling phase), the braces are both active; in the second phase (the post-buckling phase), the compressed braces are buckled; in the third phase (the plastic phase), the braces in tension plasticize. It follows that if it is necessary to perform a linear response spectra analysis, the different phases have to be appropriately taken into account. In [7], a method to obtain a trilinear pushover "spindle" was proposed, which was suitable to correctly represent the three phases.

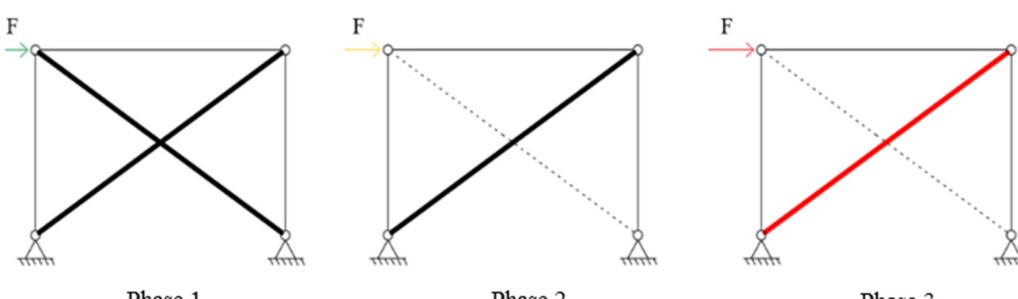

**Figure 1.** Schematization of the three phases of a CBF response under an increasing load; phase 1: elastic behavior with both diagonals active; phase 2: elastic behavior with a single active diagonal; phase 3: plastic phase.

　　　　The current EN1998-1 (also referred to as Eurocode 8 or EC8 [1]), in a seismic elastic analysis, requires us to consider the contributions of the in-tension diagonals only. This design methodology of X-CBFs steel structures may lead to uneconomical and poorly efficient structures, as underlined in several past studies [8–10]. If the designer also wants to consider the compressed members, it is necessary to perform non-linear static or dynamic analyses, which are more accurate and complete but surely more complex than the elastic ones [11].

　　　　The American AISC 341-16 [2], otherwise, also requires us to consider the compressed diagonal. Two separate analyses are requested: one elastic, in which all of the braces are assumed to resist the seismic action with their expected strength in tension or compression (pre-buckling phase), and a second one, plastic, in which the in-tension diagonal is with the expected strength, and the compressed one is with the expected post-buckling strength [2,12].

　　　　The Canadian [13] and Japanese [14] codes, in a similar way, require two different checks for the two behavior phases, with different relationships.

　　　　In this work, a simplified design method based on a linear elastic response spectrum analysis is presented. A modified stiffness of the system will be defined, as obtained by an appropriate reduction of the brace area. The modified stiffness is introduced because, if only the diagonal in tension is considered (Figure 2b), the total stiffness of the CBF would be underestimated; vice versa, if both braces are considered (Figure 2a), the total stiffness would be overestimated. Thus, the real behavior of an X-CBF in elastic conditions is intermediate to these two conditions.

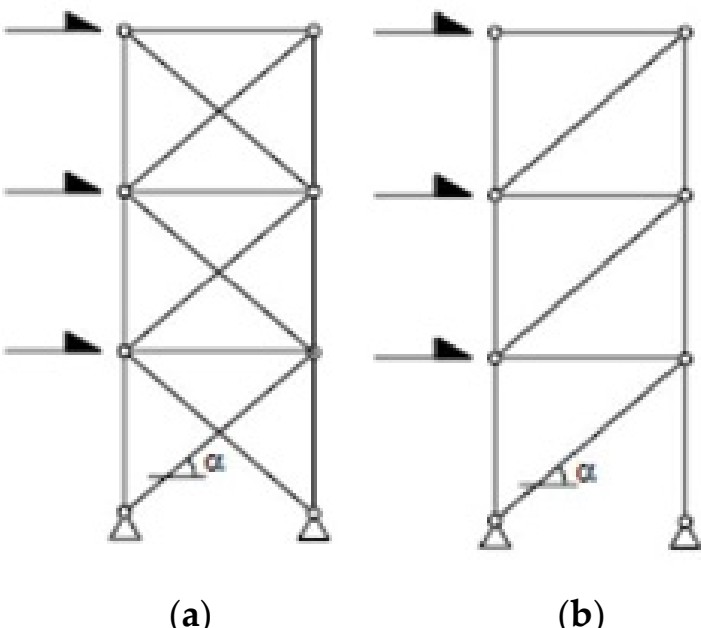

**(a)**　　　　　　　　　　　　　　　**(b)**

**Figure 2.** Modeling schemas of X-CBF according to the regulations, with both in-tension and compressed braces on the left (**a**), and with only the in-tension ones on the right (**b**).

　　　　A proper evaluation of the resultant stiffness is also important for the definition of seismic actions. In fact, considering a system such as that in Figure 3, which is made up of a seismic mass $m$ and a stiffness of the single diagonal system equal to $K_2$ (i.e., the stiffness of the second phase, Equation (1)), the ratio between the main periods of vibration of the two limit schemes is equal to a factor of $\sqrt{2}$, as can be noted by Equations (2) and (3). This clearly can give different spectral design accelerations. For simplicity, in Figure 3, a pinned beam-to-column connection is modeled. Because the axial contribution of the braces is predominant for the lateral response of the X-CBF, the results obtained could also be extended to semi-rigid joints.

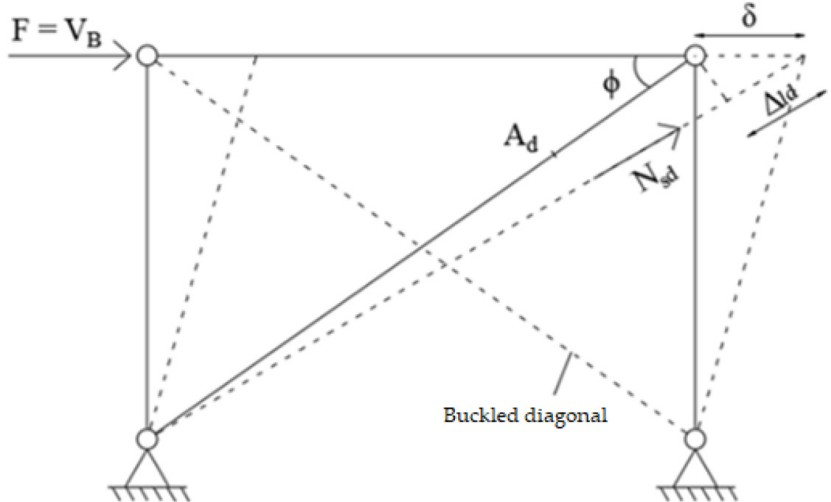

**Figure 3.** Calculation scheme for the second-phase stiffness, $K_2$.

The stiffness of the single diagonal system is:

$$K_2 = \frac{V_B}{\delta} = \frac{E \cdot A_d \cdot \cos^2 \Phi}{l_d} \tag{1}$$

with:

$V_B$: shear base force;
$\delta$: top displacement;
E: Young's modulus;
$A_d$: brace cross-section;
$\Phi$: beam–column angle;
$l_d$: brace length.

The main periods of the two limit schemas are:

$$T_2 = 2\pi \cdot \sqrt{\frac{m}{K_2}}. \tag{2}$$

$$T_1 = 2\pi \cdot \sqrt{\frac{m}{2 \cdot K_2}} = \frac{T_2}{\sqrt{2}} \tag{3}$$

We should highlight that neither of the limit schemas is correct; in the proposed methodology of analysis, both braces will be considered as active but with a reduced cross-section, in order to take into account the buckling effects. In this way, for the evaluation of the stiffness and strength, only one elastic analysis is proposed, adopting a single coherent physical model.

## 2. Definition of an Equivalent Secant Stiffness

An equivalent secant stiffness $K_{sec}$ will now be defined in order to perform a response spectrum modal analysis on an X-CBF system. It is obtainable using the lower analytical pushover curve (the "Lower-bound curve" defined in [7]) of a one-floor system (Figure 3), in which the second-phase stiffness $K_{2,LB}$ is defined with a lower value than $K_2$ (given by Equation (1)). The equivalent stiffness $K_{sec.}$ is given by imposing the intersection of an elastic line response with the design point (Figure 4) defined by the condition imposed in Equation (4):

$$V_{des} \cong 0.9 \cdot V_{pl} \tag{4}$$

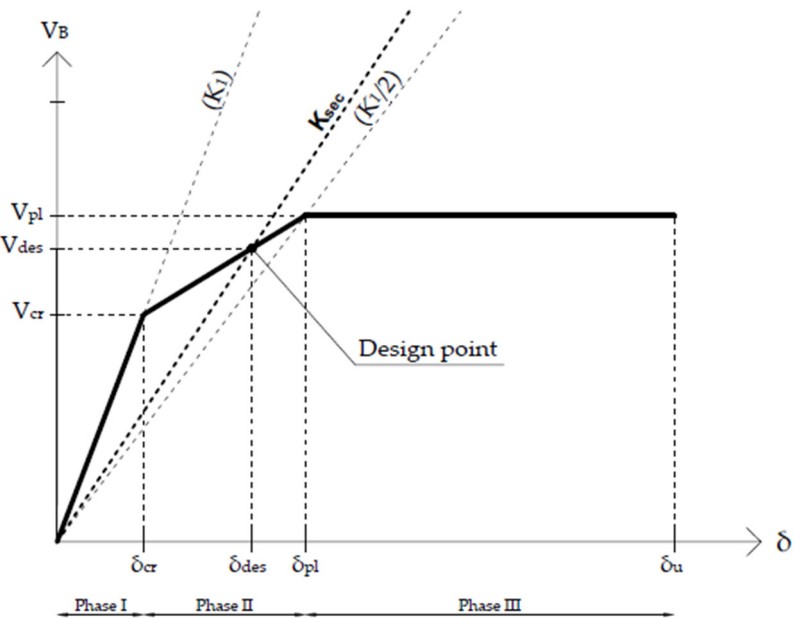

**Figure 4.** Definition of the secant $K_{sec}$ based on the Lower-bound pushover analytical curve, defined in [7].

The structure is designed on phase 2, with an elastic behavior in the tension diagonal, considering the contribution of the compressed diagonal in the buckling phase. This leads to an increase of the stiffness with respect to the condition, in which the compressed diagonal is completely neglected (EC8 design method), and an increase of the seismic actions too. The secant stiffness is estimated through the design point, which allows us to size the in-tension diagonal close to the plastic limit.

The quantities reported in Figure 4, are defined here:

$$V_{pl} = A_d \cdot f_{yd} \cdot \cos \Phi \tag{5}$$

$$V_{cr} = 2 \cdot N_{cr} \cdot \cos \Phi \tag{6}$$

$$N_{cr} = \chi(\overline{\lambda}, \alpha) \cdot \beta_A \cdot A_d \cdot f_{yd} \tag{7}$$

$$\delta_{pl} = \frac{f_{yd} \cdot l_d \cdot \beta_A}{E \cdot \cos \Phi} \tag{8}$$

$$\delta_{cr} = \frac{V_{cr}}{K_1} \tag{9}$$

$$K_1 = 2 K_2 = 2 \frac{E \cdot A_d \cdot \cos^2 \Phi}{l_d} \tag{10}$$

with

$\chi$ being the reduction coefficient for stability problems;
$\overline{\lambda}$ being the normalized slenderness of the brace;
$\alpha$ being the imperfection coefficient;
$\beta_A$ being the reduction coefficient for local effects.

The secant stiffness is thus given by:

$$K_{sec} = \frac{V_{des}}{\delta_{des}} = \frac{0.9 \, V_{pl}}{\delta_{des}} \tag{11}$$

with

$$\delta_{des} = \delta_{cr} + \frac{V_{des} - V_{cr}}{K_{2,LB}} \tag{12}$$

$$K_{2,LB} = \frac{V_{pl} - V_{cr}}{\delta_{pl} - \delta_{cr}} \tag{13}$$

The final value can be easily obtained as

$$K_{sec} = \frac{V_{des} \cdot (V_{pl} - V_{cr})}{\delta_{pl} \cdot (V_{des} - V_{cr}) + \delta_{cr} \cdot (V_{pl} - V_{des})} \tag{14}$$

Then, the percentage of correction $\Delta K$ with respect to the stiffness of the first phase is defined as

$$\Delta K(\%) = \frac{K_1 - K_{sec}}{K_1} \tag{15}$$

and by replacing

$$\Delta K\,(\%) = 1 - \frac{V_{des} \cdot \left(V_{pl} - V_{cr}\right) \cdot \delta_{cr}}{\left[\delta_{pl} \cdot (V_{des} - V_{cr}) + \delta_{cr} \cdot \left(V_{pl} - V_{des}\right)\right] \cdot V_{cr}} \tag{16}$$

This percentage of correction $\Delta K$ will be used to characterize the bracing stiffness for both mono- and multi-storey buildings.

## 3. Design Procedure

For the analysis of a generic X-CBF, four logical steps can be performed.

### 3.1. Step 1: The Pre-Design of the Single Diagonal System through an Equivalent Static Analysis

The resultant base shear of the building $V_B$ can be obtained simply using an appropriate building code, e.g., the proposal reported in §7.3.3.2 of NTC2018 [15]:

$$V_B = \frac{\lambda \cdot W \cdot S_a(T_1)}{g \cdot q} \tag{17}$$

with

W being the seismic weight;
$S_a(T_1)$ being the spectral acceleration in correspondence to the fundamental period of vibration $T_1$;
g being the gravity acceleration;
q being the behavior factor;
$\lambda$. being the coefficient equal to 0.85 if $T_1 < 2T_C$, and if the building has more than three floors, or equal to 1.0 otherwise.

Generally, the behavior factor is chosen on the basis of the performance to reach q = 1 for an elastic behavior of the structure, and q > 1 in the case of a ductile system (Figure 5). The main period $T_1$ to take into consideration for this first linear static analysis can be preliminarily determined by a modal analysis, or estimated with an approximated formula, like the one reported in Equation (18) (formula C7.3.2, given in [16]):

$$T_1 = C_1 \, H^{3/4} \tag{18}$$

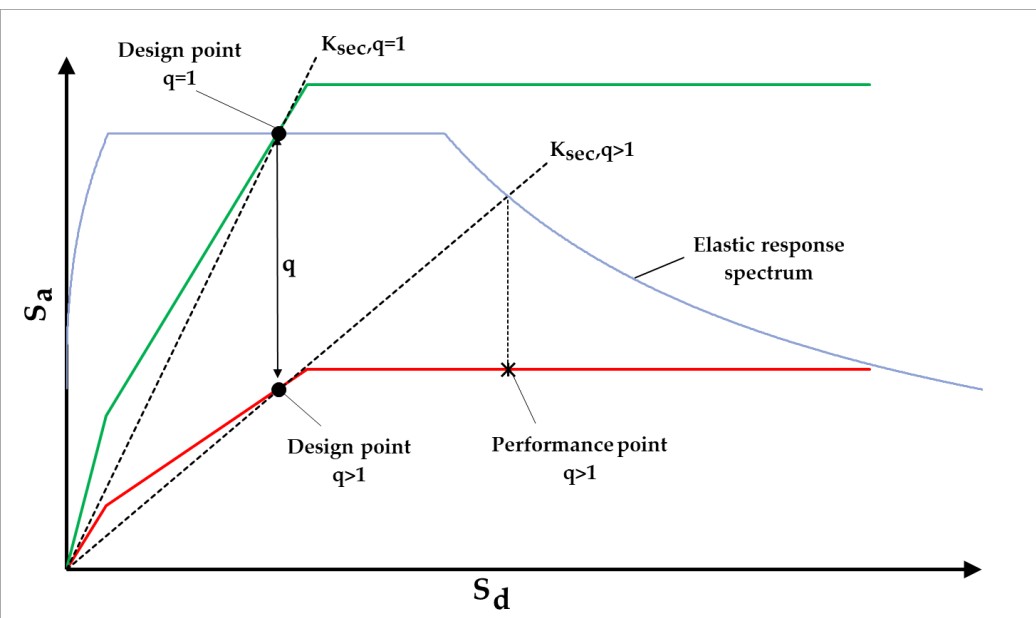

**Figure 5.** Differences between the dissipative and non-dissipative structure design methods in an Acceleration–Displacement Response Spectrum (ADRS) format chart.

The coefficient $C_1 = 0.05 \div 0.085$ (value to be used for steel structures braced or unbraced), and H is equal to the height of the building.

If a multi-storey building is analyzed, once the base shear $V_B$ is known, the shear floor $V_j$ for the j-th level must be calculated, for example, by Expression (19):

$$V_j = V_B \cdot \frac{z_j \cdot W_j}{\sum_{k=1}^{n^\circ \text{ levels}} z_k \cdot W_k} \tag{19}$$

in which

$z_j$ is the height of level j above the foundation;

$W_j$ is the seismic weight of level j.

Then, the $V_{sd,i}^{(1)}$ shear value, on the i-th X-CBF of a certain level and for the first design step, can be calculated through the distribution of floor forces and torque. If the i-th X-CBF is in the x-direction, Expression (20) gives the shear value:

$$V_{sd,i}^{(1)} = \frac{K_{x,i}}{K_{x,tot}} \cdot \left(1 \pm \frac{e_y \cdot d_{y,i}}{r_y^2}\right) \cdot V_{i,x} \tag{20}$$

with

$K_{x,i}$ being the lateral stiffness of the i-th X-CBF;

$K_{x,tot} = \sum_{k=1}^{n^\circ \text{CBF-x}} K_{x,k}$ being the total lateral stiffness given by X-CBFs in the x-direction;

$e_y$ being the eccentricity given by the structural and accidental components;

$d_{x,i}$ being the distance between the stiffness centre and the i-th X-CBF;

$r_y$ being the torsional radius, as given by Expression (21):

$$r_y = \sqrt{\frac{\sum_{k=1}^{n^\circ \text{CBF-x}} \left(K_{x,k} \cdot d_{y,k}^2\right) + \sum_{h=1}^{n^\circ \text{CBF-y}} \left(K_{y,h} \cdot d_{x,h}^2\right)}{K_{x,tot}}} \tag{21}$$

The same relationships are given if the i-th X-CBF is in the y-direction, by substituting x quantities with y ones, and vice versa.

The minimum brace area of a single diagonal X-CBF can therefore be determined by satisfying the following condition:

$$V_{sd,i}^{(1)} = V_{des} \tag{22}$$

The final area of the first design step (Figure 6) is obtained by using Equation (4):

$$A_{d,i}^{(1)} = \frac{V_{sd,i}^{(1)}}{0.9 \cdot f_{yd} \cdot \cos \Phi} \tag{23}$$

The normalized slenderness of the braces, in general, must be between 1.3 and 2, as indicated, for example, on §7.5.5 of the NTC2018 [15]. In addition, for the multi-storey buildings, the ratio $\Omega_{max}/\Omega_{min}$ between the maximum and minimum over-strength always need to be less than or equal to 25%.

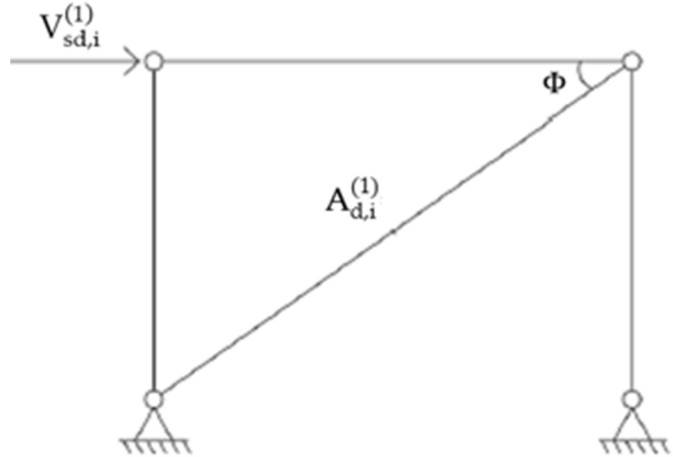

**Figure 6.** Calculation scheme of step 1.

*3.2. Step 2: Calculation of the Stiffness Correction for the Double Diagonal System*

The lateral stiffness of the single X-CBF is then calculated by considering a double-diagonal model. The equivalent secant stiffness is achievable, floor by floor, by applying the correction given by Expression (16) to the elastic double-diagonal stiffness $K_1$.

$$K_{sec} = K_1 \cdot (1 - \Delta K_i) = \frac{2 \cdot E \cdot A_{d,i}^{(1)} \cdot \cos^2 \Phi}{l_d} \cdot (1 - \Delta K_i). \tag{24}$$

The percentage correction of the stiffness can be clearly seen as the percentage correction of the brace areas. With this assumption, the corrected reduced area of the brace is calculated as follows:

$$A_{d,i}^{(2)} = A_{d,des,i}^{(1)} \cdot (1 - \Delta K_i) \tag{25}$$

*3.3. Step 3: Modal Response Spectrum Analysis with Corrected Stiffness*

In this step, the response spectrum modal analysis is performed on the double-diagonal system with reduced areas (Figure 7). The axial forces $N_{sd,i}$ on the braces are determined. Consequently, the resultant shear $V_{sd,i}^{(3)}$ on the X-CBF can be determined as

$$V_{sd,i}^{(3)} = 2 \cdot N_{sd,i} \cdot \cos \Phi \tag{26}$$

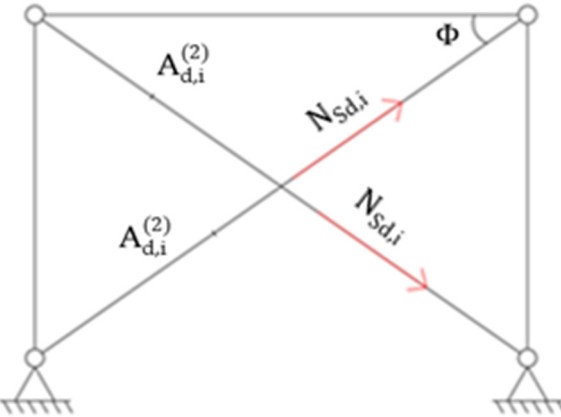

**Figure 7.** Calculation scheme of step 3.

*3.4. Step 4: Calculation of the Final Diagonal Areas*

Step 3 is necessary in order to determine the resulting design shear on the brace. With this shear force the X-CBF system has a buckled compressed diagonal. For this reason, the final diagonal area is reachable by considering the single-diagonal X-CBF with a shear equal to $V_{sd,i}^{(3)}$. (Figure 8), and by imposing the condition given by Equation (22).

$$A_{d,i}^{(4)} = \frac{V_{sd,i}^{(3)}}{0.9 \cdot f_{yd} \cdot \cos \Phi}.$$

(27)

The whole method is summarized in Table 1. For a multi-storey system, the approach based on the use of an equivalent secant stiffness can be applied floor by floor on the single X-CBF.

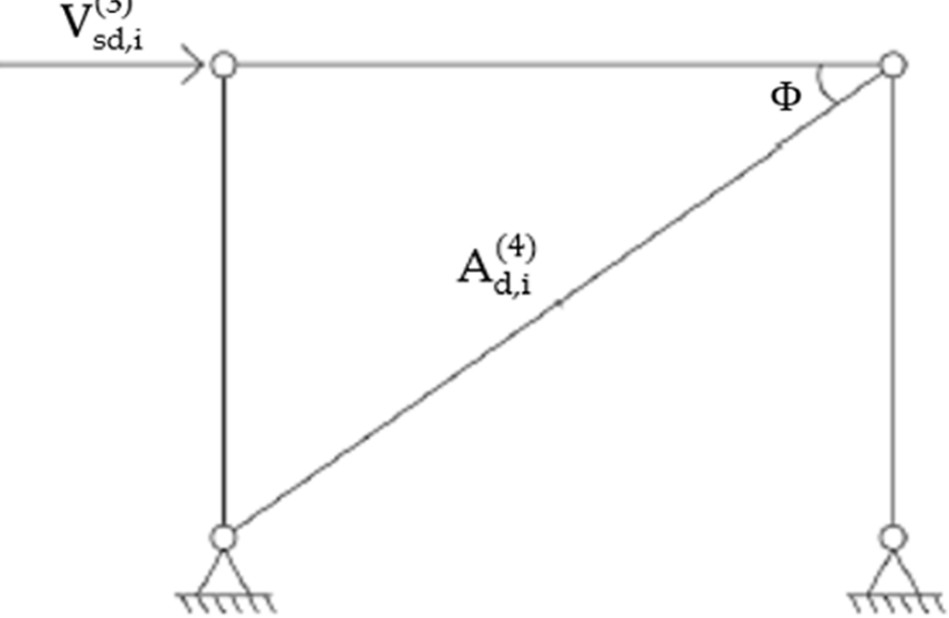

**Figure 8.** Calculation scheme of step 4.

**Table 1.** Summary of the proposed design method.

| The Design Method |
|---|

Linear static analysis:
base shear calculation $V_B$ (Equation (17))

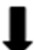

Force distribution on the floors:
shear calculation on each single X-CBF $V_{sd,i}^{(1)}$

Construction of the Analytical
Lower-Bound curve [7]

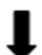

**Step 1**

Diagonal area step 1: calculation of
$A_{d,i}^{(1)}$ consequent to the imposition of
$V_{sd,i}^{(1)} = V_{des}$ (Equations (22) and (23))

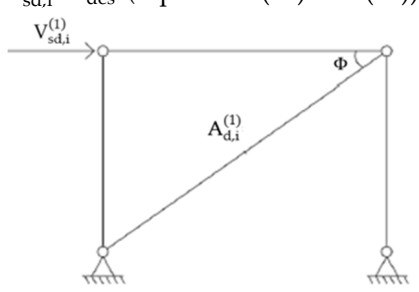

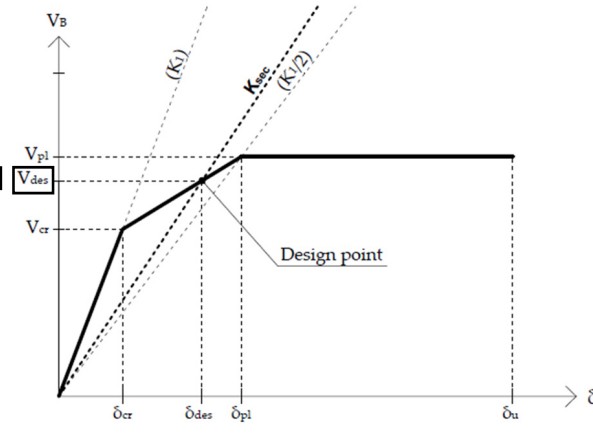

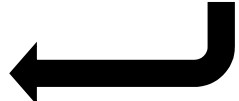

**Step 2**

Selection of the
design profile
$A_{d,des,i}^{(1)} \simeq A_{d,i}^{(1)}$

Calculation of the correction
$\Delta K_i = \Delta K_i$ ($V_{des}$, L.B. curve)
(Equation (16))

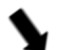

Calculation of the correct profile $A_{d,i}^{(2)}$
(Equation (25))

**Step 3**

Linear dynamic analysis on a double-diagonal
system with correct areas: calculation of the
new shear on the X-CBF
$N_{sd,i} \longrightarrow V_{sd,i}^{(3)}$ (Equation (26))

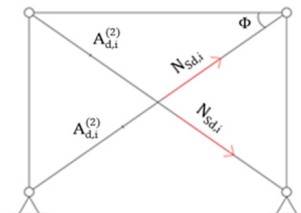

**Step 4**

Final minimum diagonal area calculation:
$A_{d,i}^{(4)}$ (Equation (27))

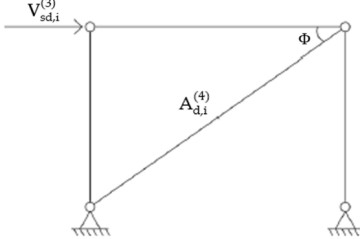

## 4. Case Studies and the Validation of the Method

The proposed method can be used to design whole mono- and multi-storey buildings. Different case studies have been analyzed, and are identifiable according to the following parameters:

- mono-storey (MONO) or multi-storey (MULTI) buildings;
- symmetrical (SYM) or not-symmetrical (NOSYM) X-CBF disposition;
- a behavior factor equal to 1 or 4.

In total, eight cases were analyzed. The structures were considered to be located in Tolmezzo (Udine, Italy), on a B class ground, of $T_1$ topographic class, with a 5% damping coefficient (according to the NTC2018 classification [15]). The nominal design life was considered equal to 50 years, and the category of use was II. The collapse limit state (SLC) was adopted as the ultimate limit state. The structure defined through the proposed method was then subjected to three artificial spectrum-compatible accelerograms generated by SIMQKE [17]. The elastic response spectrum obtained from code [15] and those obtained from artificial accelerograms are shown in Figure 9.

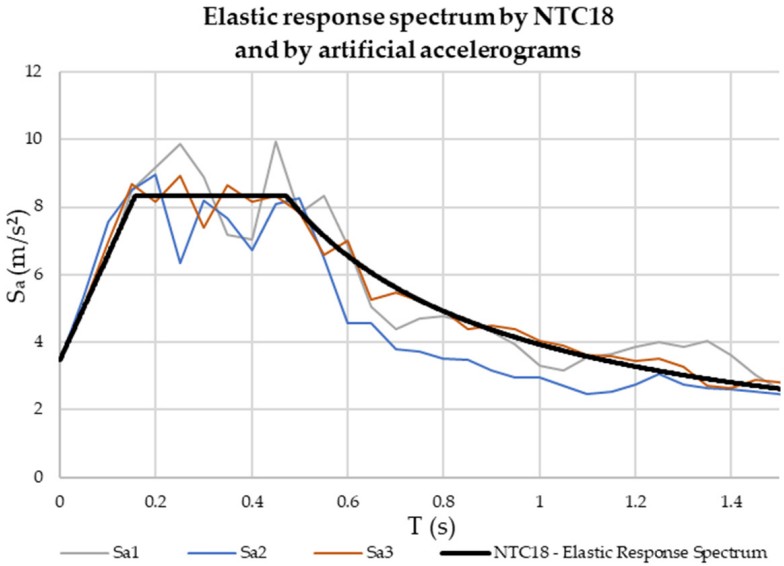

**Figure 9.** Elastic response spectrum from NTC18 and from artificial accelerograms generated by SIMQKE.

Dynamic time-histories non-linear analyses were performed with the aim to validate the design methodology. Pushover analyses were also performed, determining the target displacements, obtainable with the design elastic spectrum using the N2 method [18]. All the information of the analyses is given in single plots in the ADRS format, after the transformation of the MDOF in an SDOF system.

The description of the case studies, with the essential design steps, is reported in the next paragraphs. First, a preliminary design of the columns and beams was created by subjecting the structure to vertical loads only, with the SLU combination [15]. Then, the design of the braces was carried out with the proposed method.

### 4.1. One-Floor Building Case Studies

Some results related to a one-floor system are shown below. Every case is characterized by a single level with a height of 4 m. The loads and profiles of the beams and column are reported in Tables 2 and 3. All the case studies (also for the multi-storey buildings) are characterized by the warping of the floors and the numbering of the beams given in Figure 10.

**Table 2.** Loads applied to the one-floor case studies (from 1 to 4).

| $G_{2,\text{floor-roof}}$ | 2.50 kN/m$^2$ |
|---|---|
| $Q_{\text{accidental,roof}}$ | 1.20 kN/m$^2$ |
| $Q_{\text{snow}}$ | 1.33 kN/m$^2$ |

**Table 3.** Beam and column cross-sections of all of the one-floor case studies (from 1 to 4) after vertical static analysis design.

| Element | Profile |
|---|---|
| Beam 1 | IPE 400 |
| Beam 2 | IPE 330 |
| Beam 3 | IPE 300 |
| Beam 4 | IPE 240 |
| Column | HEB 200 |

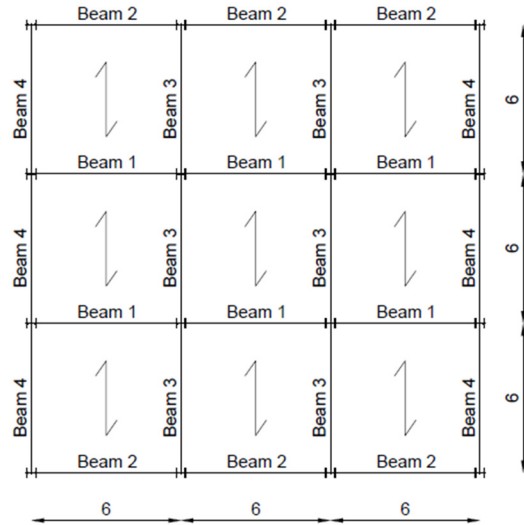

**Figure 10.** Warping of the floors and the numbering of the beams (valid for all of the case studies).

4.1.1. Case Studies 1–2: MONO–SYM q = 1 or 4

One-storey structures with a symmetric X-CBF disposition are characterized by the plan reported in Figure 11. The numerical model is shown in Figure 12. The main results are given in Tables 4 and 5.

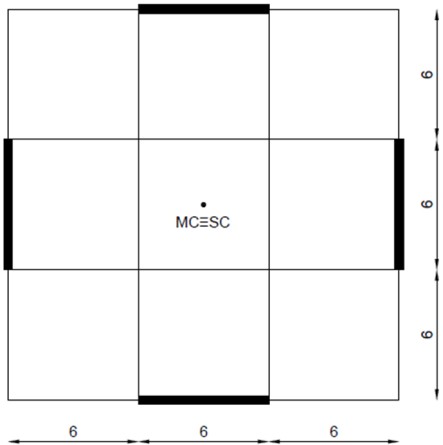

**Figure 11.** Plan of the one-floor building with the symmetrical CBF configuration (case studies 1 and 2).

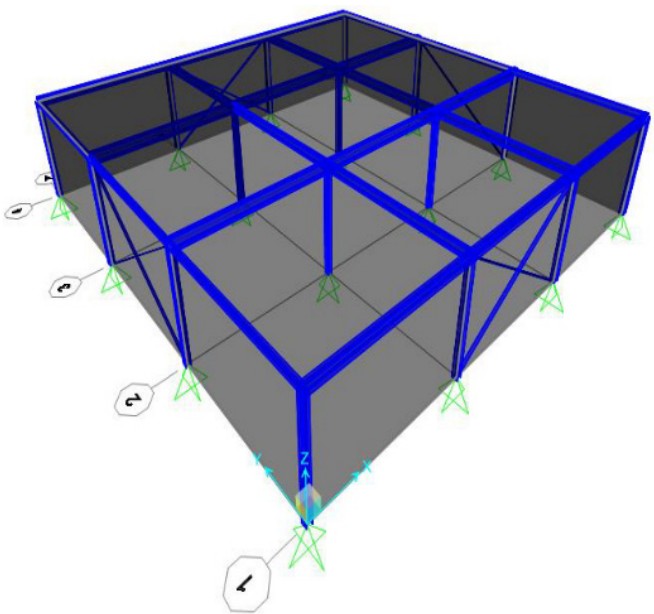

**Figure 12.** Mono-storey building model with SAP2000 for case studies 1 and 2.

**Table 4.** Results of case study 1.

| CASE 1: MONO-SYM-q = 1 | | | |
|---|---|---|---|
| **Parameters** | **Symbols** | **Values** | **Units** |
| CBF shear—step 1 | $V_{sd}^{(1)}$ | 408.5 | kN |
| Profile step 1: RHS 100 × 60 × 8 | $A_{d,des}^{(1)}$ | 2240 | mm$^2$ |
| CBF shear—step 3 | $V_{sd}^{(3)}$ | 404.16 | kN |

**Table 5.** Results of case study 2.

| CASE 2: MONO-SYM-q = 4 | | | |
|---|---|---|---|
| **Parameters** | **Symbols** | **Values** | **Units** |
| CBF shear—step 1 | $V_{sd}^{(1)}$ | 102.12 | kN |
| Profile step 1: RHS 80 × 40 × 2.5 | $A_{d,des}^{(1)}$ | 559 | mm$^2$ |
| CBF shear—step 3 | $V_{sd}^{(3)}$ | 100.93 | kN |

The method was then validated through dynamic time-history nonlinear analyses obtained with the SAP2000 code [19].

The dynamic cycles are given in plots in the ADRS format (Figures 13 and 14) after the transformation of the MDOF in an SDOF system, with pushover curves and elastic spectrum. The target displacement is obtainable as the intersection between pushover curve and elastic spectrum. In these figures, the proposed secant stiffness and the design points are also shown. In Tables 6 and 7 the maximum displacements of the time-history analyses and the target displacement of the pushover analysis are reported. In general, a good correspondence between the numerical analyses and the proposed approach was obtained using q = 1, and thus the structure remains elastic. For structures with q = 4, a response compatible with the pushover analysis was obtained, with a ductility request compatible with the capacity of the structure (Table 8).

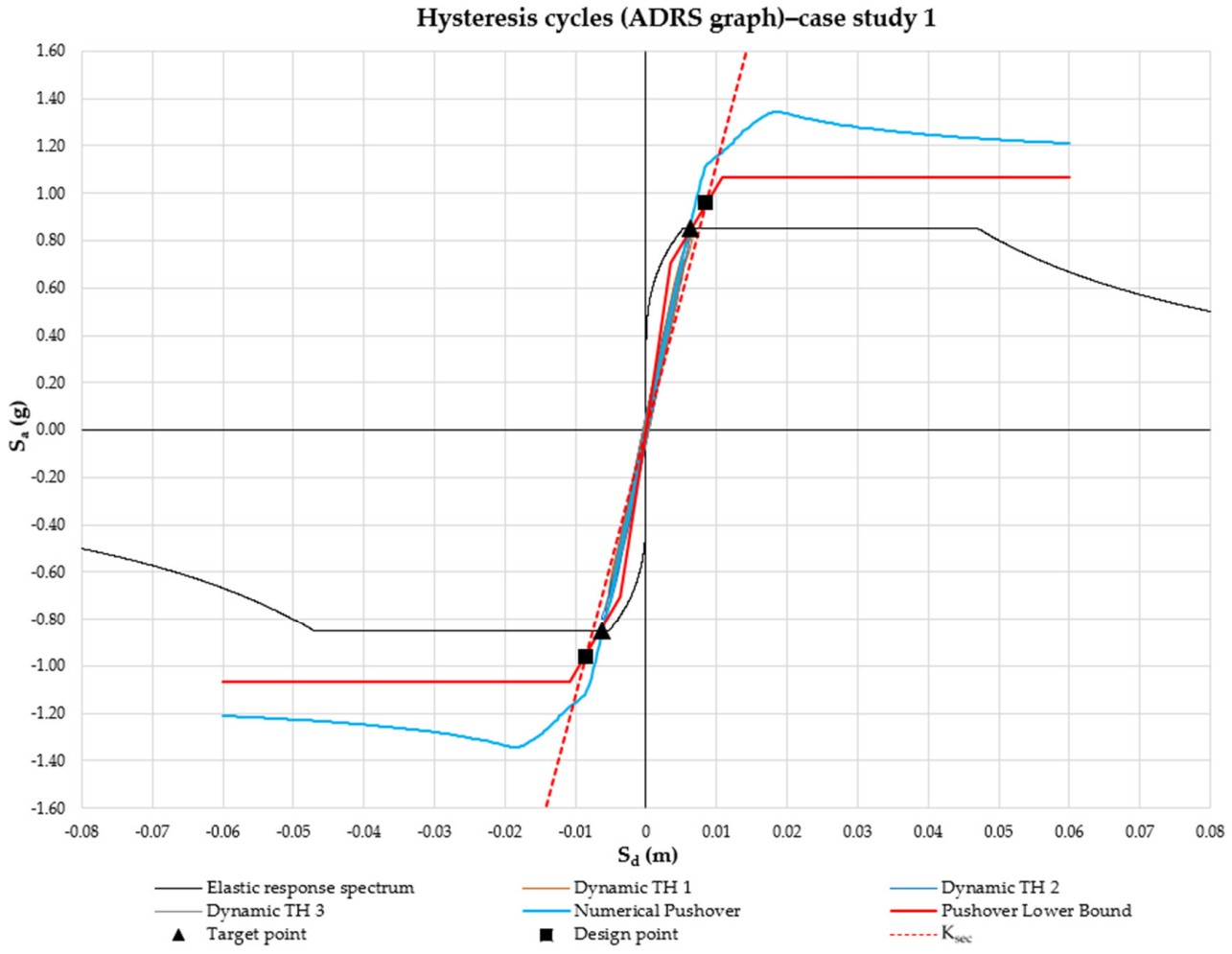

**Figure 13.** Hysteresis cycles obtained from the dynamic time-history analysis for case study 1.

**Table 6.** Maximum $S_d$ (m) for case study 1.

| | |
|---|---|
| Time history 1 | 0.007 |
| Time history 2 | 0.006 |
| Time history 3 | 0.006 |
| Target point pushover | 0.006 |

**Table 7.** Maximum $S_d$ (m) for case study 2.

| | |
|---|---|
| Time history 1 | 0.033 |
| Time history 2 | 0.029 |
| Time history 3 | 0.028 |
| Target point pushover | 0.031 |

**Table 8.** Ductility request for case study 2.

| | | |
|---|---|---|
| $T_1$ | 0.29 | s |
| $T_C$ | 0.47 | s |
| $S_{d,y}$ | 0.009 | m |
| $S_{d,max}$ | 0.033 | m |
| $\mu_d$ | 3.56 | <4 |

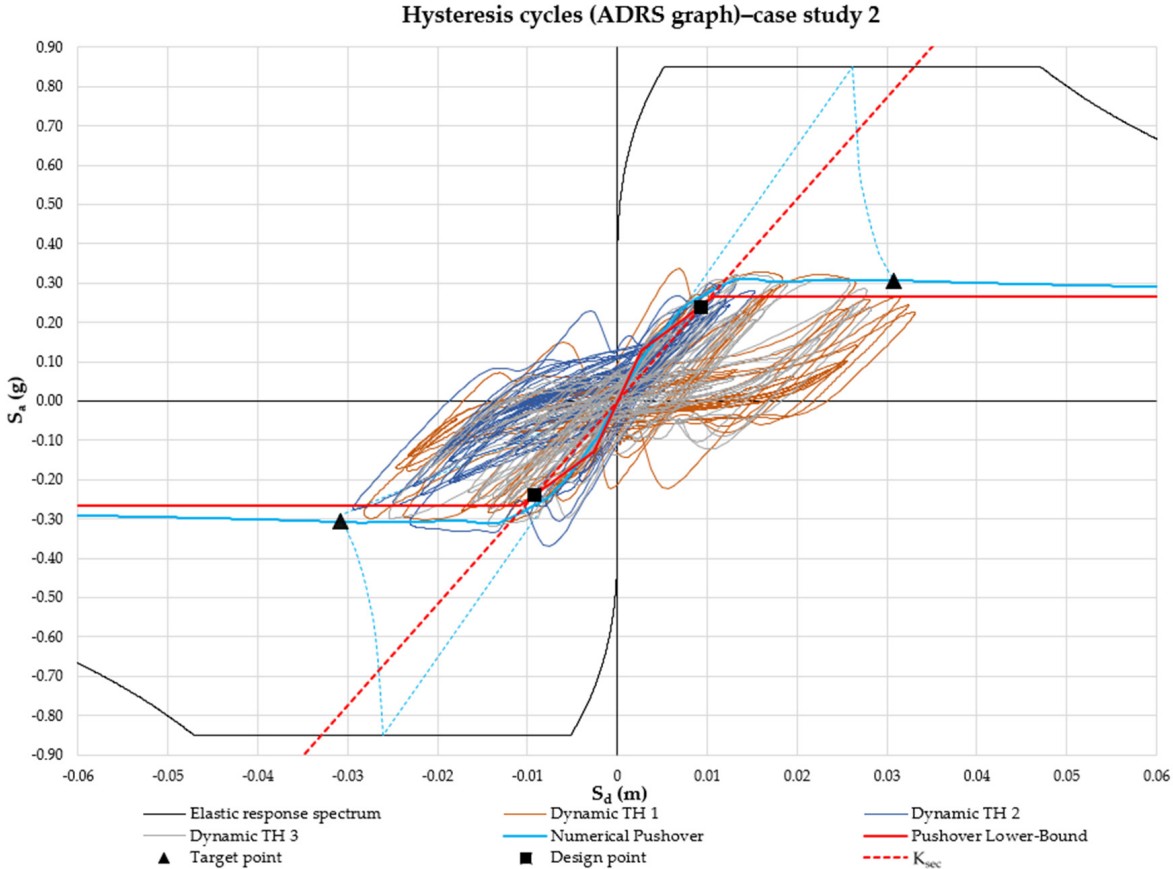

**Figure 14.** Hysteresis cycles obtained from the dynamic time-history analysis for case study 2.

### 4.1.2. Case Studies 3–4: MONO–NOSYM q = 1 or 4

Structures with a non-symmetric CBF disposition were also studied and were characterized by the plan reported in Figure 15. The external ("group 1") and internal ("group 2") areas of the CBF were distinguished. The numerical model is shown in Figure 16. The main results are given in Tables 9 and 10.

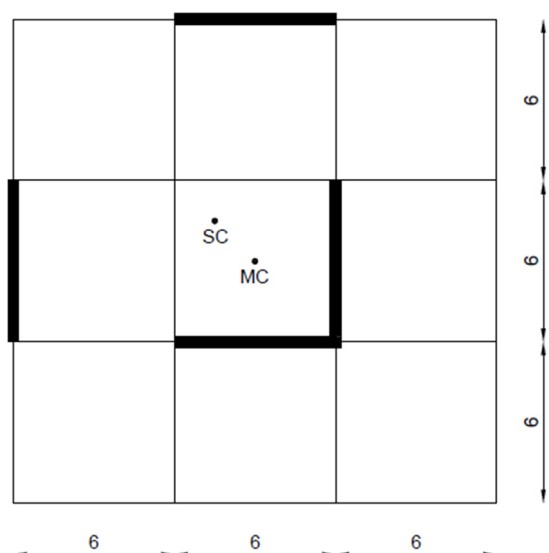

**Figure 15.** Plan of the one-floor building with the non-symmetrical CBF configuration (case studies 3 and 4).

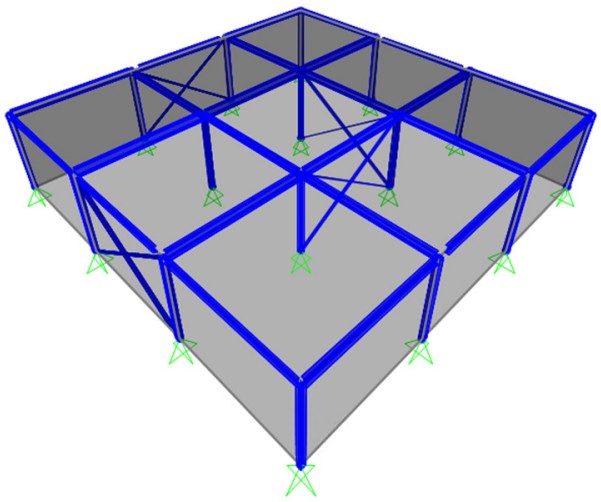

**Figure 16.** Mono-storey building model with SAP2000 for case studies 3 and 4.

**Table 9.** Results of case study 3 (G: group; S: step).

| CASE 3: MONO-NOSYM-q = 1 | | | |
|---|---|---|---|
| **Parameters** | **Symbols** | **Values** | **Units** |
| CBF shear G1—S1 | $V_{sd,G1}^{(1)}$ | 270.14 | kN |
| CBF shear G2—S1 | $V_{sd,G2}^{(1)}$ | 574.12 | kN |
| Profile G1-S1: RHS 140 × 40 × 5 | $A_{d,des,G1}^{(1)}$ | 1700 | mm$^2$ |
| Profile G2-S1: RHS 180 × 40 × 7 | $A_{d,des,G2}^{(1)}$ | 2884 | mm$^2$ |
| CBF shear G1—S3 | $V_{sd,G1}^{(3)}$ | 298.31 | kN |
| CBF shear G2—S3 | $V_{sd,G2}^{(3)}$ | 531.18 | kN |

**Table 10.** Results of case study 4 (G: group; S: step).

| CASE 4: MONO-MOSYM-q = 4 | | | |
|---|---|---|---|
| **Parameters** | **Symbols** | **Values** | **Units** |
| CBF shear G1—S1 | $V_{sd,G1}^{(1)}$ | 73.33 | kN |
| CBF shear G2—S1 | $V_{sd,G2}^{(1)}$ | 137.74 | kN |
| Profile G1—S1: RHS 60 × 40 × 2.5 | $A_{d,des,G1}^{(1)}$ | 475 | mm$^2$ |
| Profile G2—S1: RHS 80 × 40 × 3 | $A_{d,des,G2}^{(1)}$ | 684 | mm$^2$ |
| CBF shear G1—S3 | $V_{sd,G1}^{(3)}$ | 75.78 | kN |
| CBF shear G2—S3 | $V_{sd,G2}^{(3)}$ | 125.91 | kN |

### 4.1.3. Discussion on the One-Floor Building Case Studies

Generally, the shear values ($V_{sd}^{(3)}$) obtained by response spectrum analysis with a modified stiffness are very close to those obtained from the simplified static analysis ($V_{sd}^{(1)}$), as can be seen in Table 11. This is due to the fact that higher vibration modes are not significant for these cases. With the non-symmetrical structure, the values obtained with the two analyses are different because of the influence of the torsional vibration mode, which increases the shear values on the external X-CBFs.

**Table 11.** Ratios between the shears from the linear dynamic analysis with corrected stiffness, and linear static analysis for one-floor building case studies.

| Case Studies | | Group | $V_{sd}^{(3)}/V_{sd}^{(1)}$ |
|---|---|---|---|
| Symmetrical structure | q = 1 | - | 0.99 |
| | q = 4 | - | 0.99 |
| Not-symmetrical structure | q = 1 | Group 1 | 1.10 |
| | | Group 2 | 0.93 |
| | q = 4 | Group 1 | 1.03 |
| | | Group 2 | 0.91 |

### 4.2. Multi-Storey Building Case Studies

Some results related to the multi-storey case studies are shown below. Every structure is characterized by four levels: the first floor is 4 m in height, and the others are 3.5 m. The loads and profiles of the beams and columns are reported in Tables 12 and 13.

**Table 12.** Loads applied to the multi-storey case studies.

| | |
|---|---|
| $G_{2,floors}$ | 3.00 kN/m$^2$ |
| $Q_{accidental,floors}$ | 3.00 kN/m$^2$ |
| $Q_{accidental,roof}$ | 1.20 kN/m$^2$ |
| $Q_{snow}$ | 1.33 kN/m$^2$ |

**Table 13.** Beam and column cross-sections of all of the multi-storey case studies (from 5 to 8) after the vertical static analysis design.

| Element | Profile |
|---|---|
| Beam 1 | IPE 450 |
| Beam 2 | IPE 360 |
| Beam 3 | IPE 300 |
| Beam 4 | IPE 240 |
| Column | HEB 240 |

#### 4.2.1. Case Studies 5–6: MULTI–SYM q = 1 or 4

Multi-storey structures with a symmetrical X-CBF disposition are characterized by the plan reported in the Figure 17. The numerical model is shown in Figure 18. The main results are given in Tables 14 and 15.

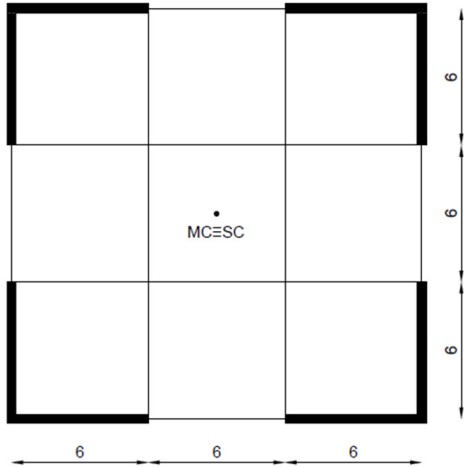

**Figure 17.** Plan of the multi-storey building with the symmetrical X-CBF configuration (case studies 5 and 6).

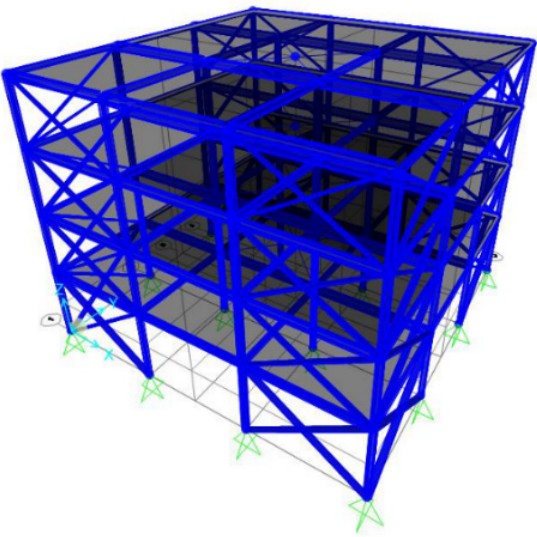

**Figure 18.** Multi-storey building model with SAP2000 for case studies 5 and 6.

**Table 14.** Results of case study 5 (L: level; S: step).

| CASE 5: MULTI-SYM-q = 1 | | | |
|---|---|---|---|
| **Parameters** | **Symbols** | **Values** | **Units** |
| Profile L1—S1: RHS 300 × 50 × 8 | $A_{d,des,L1}^{(1)}$ | 5344 mm$^2$ | mm$^2$ |
| Profile L2—S1: RHS 250 × 50 × 8,5 | $A_{d,des,L2}^{(1)}$ | 4811 mm$^2$ | mm$^2$ |
| Profile L3—S1: RHS 250 × 50 × 6 | $A_{d,des,L3}^{(1)}$ | 3456 mm$^2$ | mm$^2$ |
| Profile L4—S1: RHS 180 × 40 × 4 | $A_{d,des,L4}^{(1)}$ | 1696 mm$^2$ | mm$^2$ |
| CBF shear L1—S3 | $V_{sd,L1}^{(3)}$ | 1031.73 | kN |
| CBF shear L2—S3 | $V_{sd,L2}^{(3)}$ | 901.64 | kN |

**Table 15.** Results of case study 6 (L: level; S: step).

| CASE 6: MULTI-SYM-q = 4 | | | |
|---|---|---|---|
| **Parameters** | **Symbols** | **Values** | **Units** |
| Profile L1—S1: RHS 100 × 60 × 3 | $A_{d,des,L1}^{(1)}$ | 914 | mm$^2$ |
| Profile L2—S1: RHS 100 × 50 × 3 | $A_{d,des,L2}^{(1)}$ | 854 | mm$^2$ |
| Profile L3—S1: RHS 80 × 40 × 3 | $A_{d,des,L3}^{(1)}$ | 674 | mm$^2$ |
| Profile L4—S1: RCS 60 × 40 × 2 | $A_{d,des,L4}^{(1)}$ | 374 | mm$^2$ |
| CBF shear L1—S3 | $V_{sd,L1}^{(3)}$ | 161.78 | kN |
| CBF shear L2—S3 | $V_{sd,L2}^{(3)}$ | 144.39 | kN |

In Figures 19 and 20, as for the one-floor system, the dynamic numerical response is compared with the target point of the pushover analysis of the designed structure with the proposed approach. A good correspondence was—in general—obtained, with an elastic behaviour guaranteed for case study 5 (Table 16), and a compatible ductility request for case study 6 (Tables 17 and 18).

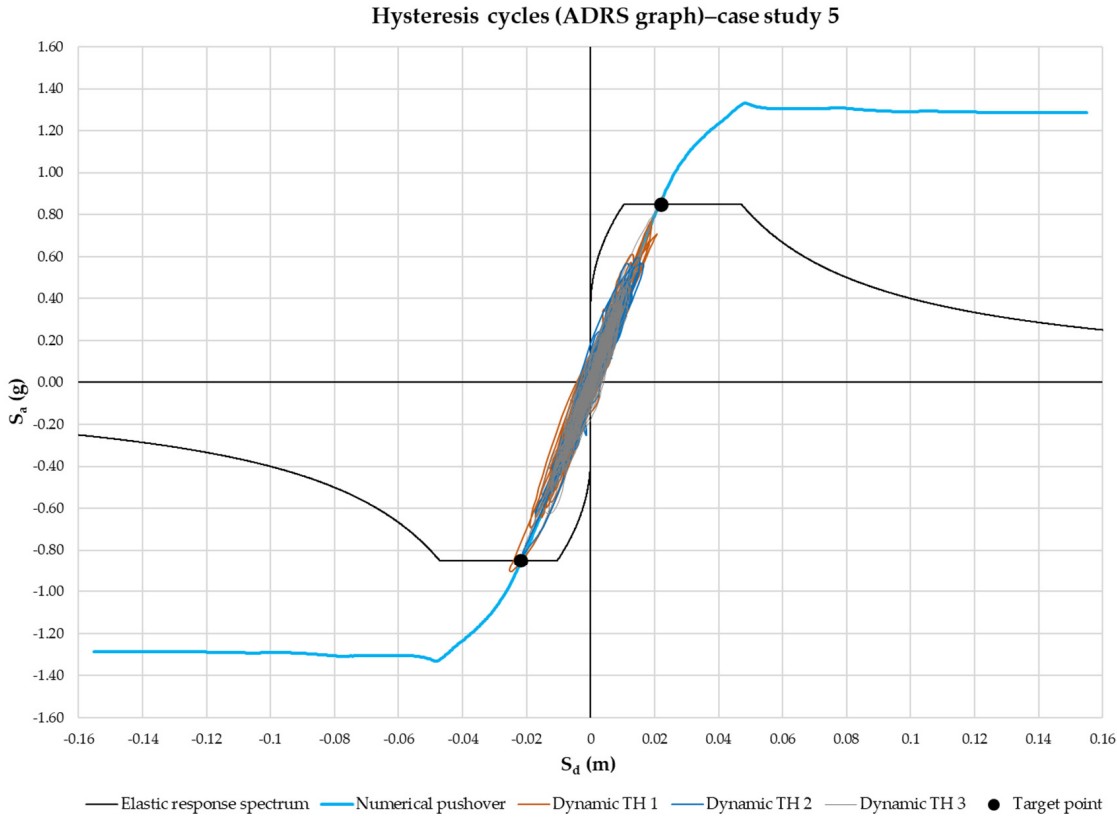

**Figure 19.** Hysteresis cycles obtained from the dynamic time-history analysis for case study 5.

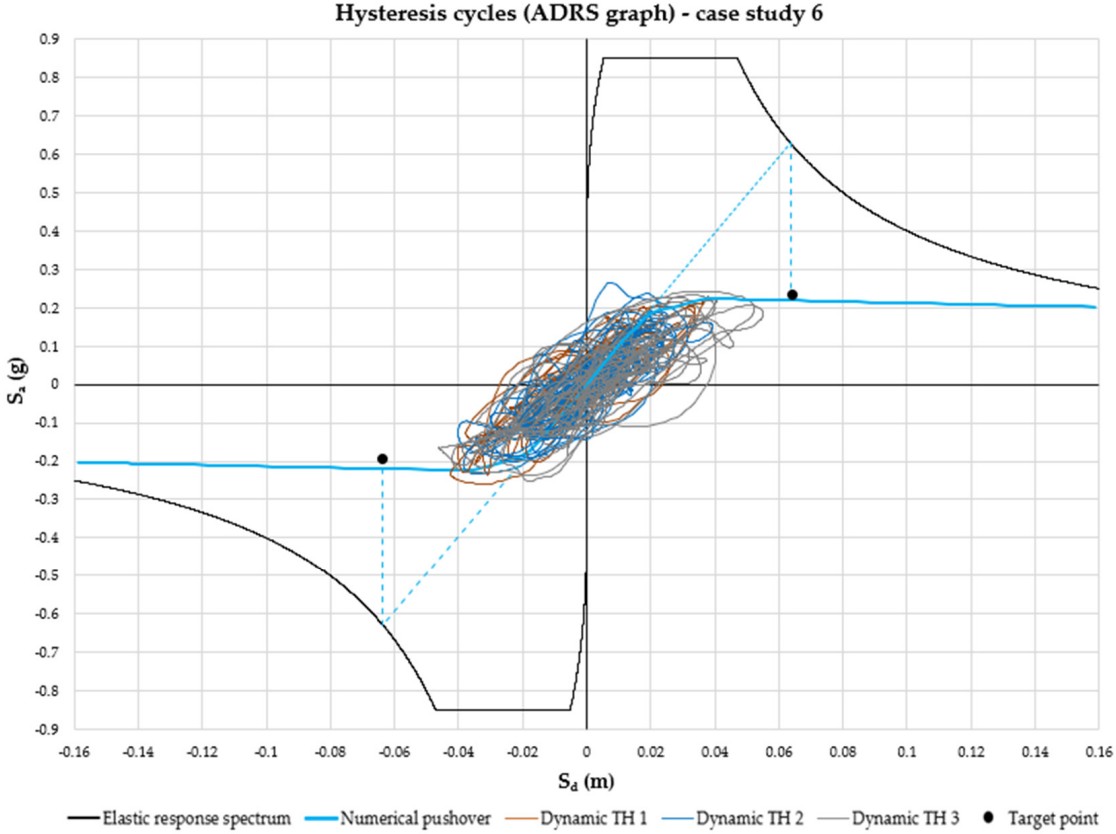

**Figure 20.** Hysteresis cycles obtained from the dynamic time-history analysis for case study 6.

**Table 16.** Maximum $S_d$ (m) for case study 5.

| | |
|---|---|
| Time history 1 | 0.025 |
| Time history 2 | 0.021 |
| Time history 3 | 0.023 |
| Target point pushover | 0.022 |

**Table 17.** Maximum $S_d$ (m) for case study 6.

| | |
|---|---|
| Time history 1 | 0.033 |
| Time history 2 | 0.029 |
| Time history 3 | 0.028 |
| Target point pushover | 0.031 |

**Table 18.** Ductility request for case study 6.

| | | |
|---|---|---|
| $T_1$ | 0.62 | s |
| $T_C$ | 0.47 | s |
| $S_{d,y}$ | 0.022 | m |
| $S_{d,max}$ | 0.055 | m |
| $\mu_d$ | 2.51 | <4 |

4.2.2. Case Studies 7–8: MULTI–NOSYM q = 1 or 4

Multi-storey structures with a non-symmetric X-CBF disposition were also studied, and were characterized by the plan reported in Figure 21. The numerical model is shown in Figure 22. The external ("group 1") and internal ("group 2) CBFs were distinguished, as in the mono-storey cases. The main results are given below (Tables 19 and 20).

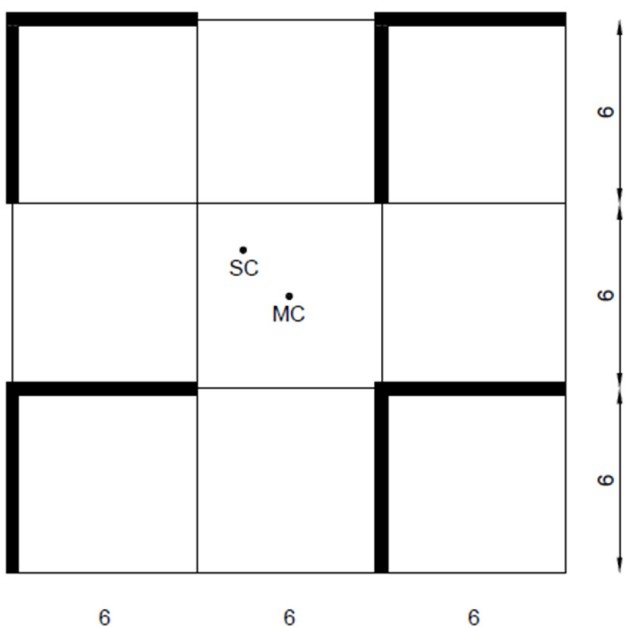

**Figure 21.** Plan of the multi-storey building with the non-symmetrical X-CBF configuration (case studies 7 and 8).

4.2.3. Discussion on the Multi-Storey Building Case Studies

As with the one-floor cases, the shear values ($V_{sd}^{(3)}$) obtained from the response spectrum analysis using the modified stiffness are close to those obtained from the simplified static analysis ($V_{sd}^{(1)}$), as can be seen in Tables 21 and 22. Moderate differences can be

observed for the dissipative cases with q = 4: the shear forces from the response spectrum analysis are, in general, slightly inferior to those of the static analysis.

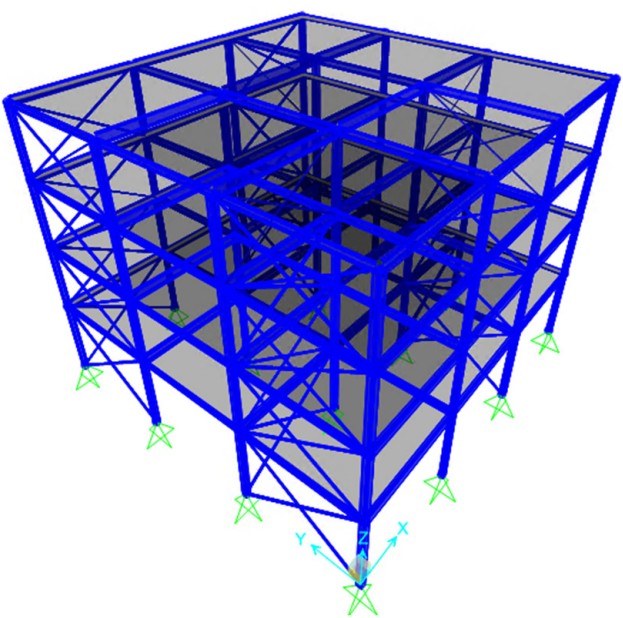

**Figure 22.** Multi-storey building model with SAP2000 for case studies 7 and 8.

**Table 19.** Results of case study 7 (L: level; G: group; S: step).

| CASE 7: MULTI-NOSYM-q = 1 | | | |
|---|---|---|---|
| **Parameters** | **Symbols** | **Values** | **Units** |
| Profile L1—G1—S1: RHS 300 × 50 × 6 | $A_{d,des,L1\text{-}G1}^{(1)}$ | 4056 | mm$^2$ |
| Profile L1—G2—S1: R 55 × 115 | $A_{d,des,L1\text{-}G2}^{(1)}$ | 6325 | mm$^2$ |
| Profile L2—G1—S1: RHS 250 × 50 × 6,3 | $A_{d,des,L2\text{-}G1}^{(1)}$ | 3621 | mm$^2$ |
| Profile L2—G2—S1: R 55 × 100 | $A_{d,des,L2\text{-}G2}^{(1)}$ | 6000 | mm$^2$ |
| Profile L3—G1—S1: RHS 250 × 50 × 5 | $A_{d,des,L3\text{-}G1}^{(1)}$ | 2900 | mm$^2$ |
| Profile L3—G2—S1: RHS 300 × 50 × 6,3 | $A_{d,des,L3\text{-}G2}^{(1)}$ | 4251 | mm$^2$ |
| Profile L4—G1—S1: RHS 130 × 50 × 4 | $A_{d,des,L4\text{-}G1}^{(1)}$ | 1376 | mm$^2$ |
| Profile L4—G2—S1: RHS 150 × 50 × 6 | $A_{d,des,L4\text{-}G2}^{(1)}$ | 2256 | mm$^2$ |
| Profile L1—G1—S1: RHS 300 × 50 × 6 | $A_{d,des,L1\text{-}G1}^{(1)}$ | 4056 | mm$^2$ |
| Profile L1—G2—S1: R 55 × 115 | $A_{d,des,L1\text{-}G2}^{(1)}$ | 6325 | mm$^2$ |

**Table 20.** Results of case study 8 (L: level; G: group; S: step).

| CASE 8: MULTI-NOSYM-q = 4 | | | |
|---|---|---|---|
| **Parameters** | **Symbols** | **Values** | **Units** |
| Profile L1—G1—S1: RHS 60 × 40 × 4 | $A_{d,des,L1-G1}^{(1)}$ | 736 | mm$^2$ |
| Profile L1—G2—S1: RHS 80 × 40 × 6 | $A_{d,des,L1-G2}^{(1)}$ | 1296 | mm$^2$ |
| Profile L2—G1—S1: RCS 60 × 40 × 4 | $A_{d,des,L2-G1}^{(1)}$ | 736 | mm$^2$ |
| Profile L2—G2—S1: RHS 60 × 40 × 6 | $A_{d,des,L2-G2}^{(1)}$ | 1056 | mm$^2$ |
| Profile L3—G1—S1: RCS 60 × 40 × 3 | $A_{d,des,L3-G1}^{(1)}$ | 564 | mm$^2$ |
| Profile L3—G2—S1: RHS 300 × 50 × 6,3 | $A_{d,des,L3-G2}^{(1)}$ | 804 | mm$^2$ |
| Profile L4—G1—S1: RCS 50 × 30 × 2 | $A_{d,des,L4-G1}^{(1)}$ | 304 | mm$^2$ |
| Profile L4—G2—S1: CHS 60,3 × 2,6 | $A_{d,des,L4-G2}^{(1)}$ | 471.3 | mm$^2$ |
| Profile L1—G1—S1: RHS 60 × 40 × 4 | $A_{d,des,L1-G1}^{(1)}$ | 736 | mm$^2$ |
| Profile L1—G2—S1: RHS 80 × 40 × 6 | $A_{d,des,L1-G2}^{(1)}$ | 1296 | mm$^2$ |

**Table 21.** Ratios between the shears from the linear dynamic analysis with corrected stiffness and the linear static analysis for all of the multi-storey building case studies with the symmetrical disposition of the X-CBFs.

| Ratio $V_{sd}^{(3)}/V_{sd}^{(1)}$ Symmetrical Structure | | |
|---|---|---|
| **Floor** | **q = 1** | **q = 4** |
| Level 1 | 1.01 | 0.87 |
| Level 2 | 1.00 | 0.88 |
| Level 3 | 1.04 | 0.89 |
| Level 4 | 1.09 | 1.01 |

**Table 22.** Ratios between the shears from the linear dynamic analysis with corrected stiffness and the linear static analysis for all of the multi-storey building case studies with the non-symmetrical disposition of the X-CBFs.

| Floor | Ratio $V_{sd}^{(3)}/V_{sd}^{(1)}$ Not Symmetrical Structure | | | |
|---|---|---|---|---|
| | **q = 1** | | **q = 4** | |
| | **G1** | **G2** | **G1** | **G2** |
| Level 1 | 1.36 | 1.36 | 0.73 | 0.93 |
| Level 2 | 1.29 | 0.88 | 0.83 | 0.84 |
| Level 3 | 1.43 | 0.97 | 0.96 | 0.88 |
| Level 4 | 1.41 | 1.11 | 0.90 | 1.07 |

With q = 1, for a symmetrical configuration, the results of the modal analyses were similar to those of the static analysis, while for a non-symmetrical system there were differences because of the participation of the torsional vibration modes.

With q = 4, for both the symmetrical and non-symmetrical configurations, the shear values obtained with modal analysis were minor compared to those with the static analysis. This is principally due to the periods considered in the static analysis that are calculated

on the basis of an empirical relationship. This leads to period shorter than the real ones obtained with the modal analysis.

## 5. Conclusions

In this paper, the behavior and the design of concentric X-braced steel frames was analyzed.

It was initially pointed out that many of the current building codes consider only the "pre-buckling" or "post-buckling" phases in order to elastically model the structure. If only the in-tension brace is considered, the X-CBF stiffness is—in general—under-evaluated; on the contrary, by considering the compressed member, the stiffness is in general over-evaluated. The actual behaviour is in fact intermediate to these two limit situations.

A new approach for the design of the X-CBFs, based on the use of a modal analysis with a response spectrum, using an appropriate modified stiffness of braces, was presented. A series of case studies were analyzed, using the proposed method. The proposal was validated through non-linear dynamic analyses carried out using artificial spectrum-compatible accelerograms. The method demonstrated its effectiveness for one-floor or multi-storey buildings, with a dissipative or not dissipative behaviour, with a symmetrical or non-symmetrical X-CBF position. The proposed approach could therefore be very important to the performance of a correct modal analysis and the design of X-CBF systems with active tension diagonal bracings using the response spectrum method. The approach is unified, coherent with the physical behaviour, and valid for slender or squat diagonal bracings.

**Author Contributions:** Conceptualization, C.A. and L.B.; methodology, C.A. and L.B.; software, L.B.; validation, C.A., L.B. and S.N.; formal analysis, C.A. and L.B.; investigation, C.A. and L.B.; resources, C.A., L.B. and S.N.; data curation, L.B. and S.N.; writing—original draft preparation, L.B.; writing—review and editing, C.A., L.B. and S.N.; visualization, C.A., L.B. and S.N.; supervision, C.A., L.B. and S.N.; project administration, C.A., L.B. and S.N. All authors have read and agreed to the published version of the manuscript.

**Funding:** This research was funded by DPC-ReLUIS project 2019–2021 (WP12), Italy.

**Data Availability Statement:** The authors have all the data.

**Conflicts of Interest:** The authors declare no conflict of interest.

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
