# Peer review of "Design of X-Concentric Braced Steel Frame Systems Using an Equivalent Stiffness in a Modal Elastic Analysis"

_buildings, doi:10.3390/buildings12030359_

Round 1
Reviewer 1 Report
This paper presents a numerical study for the seismic response of CBF steel frames and considers the arrangement of diagonal bracings. A series of modal response spectrum analyses are performed with looking at buckling behaviour of braces. A new approach is presented using modal analysis.
I suggest publication after minor amendments. The Authors may wish to consider the following points:
Technical:
1. L110: how this equivalent stiffness compares with the normal stiffness used for dynamic analysis?
Editorial:
- L40, L77 etc.: check reference.
- L304: check table alignment
Author Response
Response to Reviewer 1 Comments
Technical:
- L110: how this equivalent stiffness compares with the normal stiffness used for dynamic analysis?
Ksec is the stiffness proposed for the dynamic modal analysis. The comparison between this and the one that consider both diagonal (K1) is reported in the relationships (15) and (16). The stiffness recommended by EC8 is the one obtainable by only considering the in tension diagonal, that is K1/2.
Editorial:
- L40, L77 etc.: check reference.
OK
- L304: check table alignment
OK
Please see the attachment for the new, modified version of the article. All the variations are reported in red text.

Reviewer 2 Report
The paper concerns the method for the design of concentric braced steel frames with active tension diagonal bracings using modal elastic analysis. It was presented for both single-storey and multi-storey buildings. The paper is written at a good level. Nevertheless, it could be improved according to the following guidelines:
Major revisions
- Line 28. The literature review must be broader than that presented in the paper.
- Add a discussion of the results at the end of Sections 4.1 and 4.2.
- Add a quantitative analysis for the pairs of plots in Figures 13 and 14, and Figures 19 and 20.
Minor revisions
- The table titles for all tables in the paper should be moved above the tables.
- Lines: 40, 77, 126. Delete any misspelled text.
- Page 15. Remove duplicate Tables 13 and 14.
Author Response
Response to Reviewer 2 Comments
Please see the attachment for the new, modified version of the article. All the variations are reported in red text.
Major revisions
1. Line 28. The literature review must be broader than that presented in the paper.
We agree with the reviewer’s request. The first part of the introduction (and the bibliography too), that concerns the literature review, has been reviewed and implemented (lines 18-43, page 1):
Concentric braced steel frames with active tension diagonal bracings (X-CBF) represent one of the most used structural types to withstand earthquakes or wind forces. They are widely used for both mono and multi-storey buildings due to the high dissipative capacity and cheapness. During a seismic event, the energy is dissipated through the diagonals, that plasticize in tension and buckle in compression. The other parts, like beams and columns, are generally designed to remain elastic [1]–[3].
Many experimental and numerical studies, as in [4]–[6], have demonstrated that a X-CBF subjected to increasing horizontal actions is characterized by a three-phases behaviour (Figure 1). In the first phase (“Pre-Buckling phase”) the braces are both active; in the second one (“Post-Buckling phase”) the compressed braces is buckled; in the third one (“Plastic phase”) the braces in tension plasticizes. It follows that if it is necessary to perform a linear response spectra analysis, the different phases have to be appropriately taken into account. In [7] a method to get a trilinear pushover “spindle” has been proposed, suitable to correctly represent the three phases.
Current EN1998-1 (also referred as Eurocode 8 or EC8, [1]), in a seismic elastic analysis, requires to consider the contributions of the in tension diagonals only. This design methodology of X-CBFs steel structures may lead to uneconomical and poorly efficient structures, as underlined in several past studies [8]–[10]. If the designer wants also to consider the compressed members, it is necessary to perform non-linear static or dynamic analyses, more accurate and complete but surely more complex than the elastic ones [11].
The American AISC 341-16 [2], otherwise, requires also to consider the compressed diagonal. Two separate analyses are requested: one elastic in which all braces are assumed to resist to the seismic action with their expected strength in tension or compression (pre-buckling phase) and a second one, plastic, in which the in tension diagonal is with his expected strength and the compressed one with his expected post-buckling strength [2], [12].
The Canadian [13] and Japanese [14] codes, in a similar way, require two different checks for the two behavior phases, with different relationships.
2. Add a discussion of the results at the end of Sections 4.1 and 4.2.
There are two paragraphs about observations on the one floor building (4.1.3) and multi-storey (4.2.3) case studies. Some additional remarks have been added:
Chapter 4.1.3 (lines 290-292; page 11):
With not-symmetrical structure, the values obtained with the two analyses are more different because of the influence of the torsional vibration mode, which increases shear values on the external X-CBFs.
Chapter 4.2.3 (lines 303-305; page 12):
With q=1 and for a symmetrical configuration, the results of the modal analysis are similar to those with the static analysis, while for a not symmetrical configuration there is a difference because of the participation of the torsional vibration modes. With q=4, for both symmetrical and not-symmetrical configuration, shear values obtained with modal analysis are minor than those with the static analysis. This is principally due to the periods considered in the static analysis that are calculated on the basis of an empirical relationship. This leads to period shorter than the real ones, obtained with the modal analysis.
3. Add a quantitative analysis for the pairs of plots in Figures 13 and 14, and Figures 19 and 20.
Table 6-7-8-16-17-18 are added, in order to give some quantitative results, that consist in the maximum spectral displacements for every case, and for the ductility request for the cases with q=4.
Minor revisions
1. The table titles for all tables in the paper should be moved above the tables.
OK
2. Lines: 40, 77, 126. Delete any misspelled text.
OK
3. Page 15. Remove duplicate Tables 13 and 14.
OK

Reviewer 3 Report
This paper presented the design of concentrically braced steel frames (CBFs) through theoretical derivation and numerical validation, in which the contribution of buckling bracings on the lateral stiffness has been taken into account. Some problems listed below should be modified or explained accordingly.
- My major concern of this paper is the research purpose. The introduction may not provide sufficient background about the existing design methods of ordinary CBFs, especially the methods in America, Japan, and China, which differ from the Eurocode method. The contribution of bracings in compression may not be neglected in other existing methods. Therefore, the reviewer doubts the innovation of this paper.
- Another one is about the CBFs in this paper featured pinned beam-to-column connections, which seems to be a special case. For CBFs with rigid or semi-rigid connections, further explanations should be added.
- The title should be shortened and more to the point without abbreviating CBF.
- Line 40. Some references failed to display. This remark does also apply to other references.
- Figure 1. More details should be added to this figure concerning the different colors and line types.
- Line 77 and Equation (4). More explanations should be given about the determination of the design point. Why does this point locate at the second phase but not the first one?
- Equation (11) and Figure 4. The secant stiffness may not be stable. Although a design shear force can be obtained, the seismic forces are actually uncertain. The actual stiffness may be lower than Ksec, likely to cause unsafe results. But the use of K2 in design will provide a considerable margin of safety.
- Line 126. Please check the word “ne”.
- Figure 5. The full name of ADRS is required to be given.
- Line 313. Please check the section number.
Author Response
Response to Reviewer 3 Comments
Please see the attachment for the new, modified version of the article. All the variations are reported in red text.
This paper presented the design of concentrically braced steel frames (CBFs) through theoretical derivation and numerical validation, in which the contribution of buckling bracings on the lateral stiffness has been taken into account. Some problems listed below should be modified or explained accordingly.
- My major concern of this paper is the research purpose. The introduction may not provide sufficient background about the existing design methods of ordinary CBFs, especially the methods in America, Japan, and China, which differ from the Eurocode method. The contribution of bracings in compression may not be neglected in other existing methods. Therefore, the reviewer doubts the innovation of this paper.
We agree with the reviewer’s request. The first part of the introduction (and the bibliography has been reviewed and implemented (lines 18-43, page 1):
Concentric braced steel frames with active tension diagonal bracings (X-CBF) represent one of the most used structural types to withstand earthquakes or wind forces. They are widely used for both mono and multi-storey buildings due to the high dissipative capacity and cheapness. During a seismic event, the energy is dissipated through the diagonals, that plasticize in tension and buckle in compression. The other parts, like beams and columns, are generally designed to remain elastic [1]–[3].
Many experimental and numerical studies, as in [4]–[6], have demonstrated that a X-CBF subjected to increasing horizontal actions is characterized by a three-phases behaviour (Figure 1). In the first phase (“Pre-Buckling phase”) the braces are both active; in the second one (“Post-Buckling phase”) the compressed braces is buckled; in the third one (“Plastic phase”) the braces in tension plasticizes. It follows that if it is necessary to perform a linear response spectra analysis, the different phases have to be appropriately taken into account. In [7] a method to get a trilinear pushover “spindle” has been proposed, suitable to correctly represent the three phases.
Current EN1998-1 (also referred as Eurocode 8 or EC8, [1]), in a seismic elastic analysis, requires to consider the contributions of the in tension diagonals only. This design methodology of X-CBFs steel structures may lead to uneconomical and poorly efficient structures, as underlined in several past studies [8]–[10]. If the designer wants also to consider the compressed members, it is necessary to perform non-linear static or dynamic analyses, more accurate and complete but surely more complex than the elastic ones [11].
The American AISC 341-16 [2], otherwise, requires also to consider the compressed diagonal. Two separate analyses are requested: one elastic in which all braces are assumed to resist to the seismic action with their expected strength in tension or compression (pre-buckling phase) and a second one, plastic, in which the in tension diagonal is with his expected strength and the compressed one with his expected post-buckling strength [2], [12].
The Canadian [13] and Japanese [14] codes, in a similar way, require two different checks for the two behavior phases, with different relationships.
We propose a novel design method because it’s different from both the typology codes (EC8 at one side and American, Japanese and Canadian on the other): we consider one elastic analysis only but keeping into account in a proper way the compressed brace. For this reason we have added the next sentence at lines 77-79, page 3:
In this way, for the evaluation of stiffness and strength, only one elastic analysis is proposed by adopting a single coherent physical model.
- Another one is about the CBFs in this paper featured pinned beam-to-column connections, which seems to be a special case. For CBFs with rigid or semi-rigid connections, further explanations should be added.
The observation has been received and introduced at lines 58-60, page 2:
For simplicity, in Figure 3, a pinned beam-to-column connection is modeled. Since the axial contribute of the braces is predominant for the lateral response of the X-CBF, the results obtained could also be extended to semi-rigid joints.
- The title should be shortened and more to the point without abbreviating CBF.
The title has changed as follows:
Design of X - concentric braced steel frames systems using an equivalent stiffness in a modal elastic analysis
- Line 40. Some references failed to display. This remark does also apply to other references.
OK
- Figure 1. More details should be added to this figure concerning the different colors and line types.
The caption has been changed:
Schematization of the three phases of a CBF response under increasing load; phase 1: elastic behavior with both diagonals active; phase 2: elastic behavior with a single active diagonal; phase 3: plastic phase.
- Line 77 and Equation (4). More explanations should be given about the determination of the design point. Why does this point locate at the second phase but not the first one?
More explanations have been added, at lines 93-98, page 3:
The structure is so designed on phase 2, with an elastic behavior and considering the contribute of the compressed diagonal in the buckling phase. This leads to an increase of the stiffness with respect to the condition in which the compression diagonal is completely neglected (EC8 design method), and an increase of the seismic actions too. The secant stiffness is estimated through the design point, which allows to size the tension diagonal close to the plastic limit.
- Equation (11) and Figure 4. The secant stiffness may not be stable. Although a design shear force can be obtained, the seismic forces are actually uncertain. The actual stiffness may be lower than Ksec, likely to cause unsafe results. But the use of K2 in design will provide a considerable margin of safety.
The secant stiffness has resulted stable in all our analyzed systems. The use of the K2 stiffness implies higher vibration periods and so lower seismic actions in an elastic analysis.
- Line 126. Please check the word “ne”.
OK
- Figure 5. The full name of ADRS is required to be given.
OK
- Line 313. Please check the section number.
OK

Round 2
Reviewer 2 Report
No further comments.
Reviewer 3 Report
The reviewer thanks the authors for checking and improving this paper carefully. All my comments have been addressed satisfactorily and this paper can be accepted for publication.